# Intermittency of Rock Fractured Surfaces: A Power Law

Saeed Aligholi * and Manoj Khandelwal *

Institute of Innovation, Science and Sustainability, Federation University Australia, Ballarat, VIC 3350, Australia
* Correspondence: s.aligholi@federation.edu.au (S.A.); m.khandelwal@federation.edu.au (M.K.)

**Abstract:** Roughness of rock fractured surfaces is one of the most important factors controlling fluid flow in rock masses. Roughness quantification is of prime importance for modelling the flow of ground waters as well as reservoir fluid mechanics. In this study, with the aid of high-resolution 3D X-ray CT scanning and image processing techniques, the roughness of four different rock types is reconstructed with a resolution of 16.5 microns. Moreover, the correlation and structure functions are used to analyse height fluctuations as well as statistical intermittency of the studied rock fractured surfaces. It is observed that at length scales smaller than a critical length scale, fractures surfaces are correlated and show multifractality. Monofractals are neither intermittent nor correlated; hence, a meaningful link between statistical intermittency and the correlation function of multifractals is expected. However, a model that considers this relationship and predicts multifractal spectra of disordered systems is still missing. A simple power law that can exactly forecast the multiscaling spectrum of rock fracture process zone is being introduced. It is explained how the exponent of this power function $\lambda_i$ is related to the crossover length of correlation function $\xi$, and how this critical length scale can be objectively identified.

**Keywords:** roughness; multifractal; rock fracture; intermittency

## 1. Introduction

Quantitative roughness analysis of rock materials is not only the key to understand mysteries associated with nonlinear inelastic fracture mechanics at small enough length scales, but also a powerful and applicable tool in many rock engineering fields [1–3].

The main goal of this work is to establish a relationship between the correlation and intermittency of rock fractured surfaces. This link not only broadens our horizons about the statistical physics of multifractal phenomena but also is a promising approach in order to objectively determine the crossover length (or time) $\xi$ of physical phenomena. $\xi$ is a very important parameter in studying the phase transition of natural phenomena [4]. If a fracture is considered as a phase transition [5], $\xi$ is a cross-over length scale that shows a transition from continuous damage percolation at $\delta r \ll \xi$ to the first order at $\delta r \gg \xi$ where material can be considered as linear elastic [6], where $\delta r$ denotes separation or length scale. This phase transition is not very sharp in quasi-brittle rock materials and takes place in a range of length scales because of the mixed first-order and continuous character [7]. Therefore, determining the true $\xi$ as the statistical critical point at which the phase transition takes place, which is reminiscent of the effective length of the fracture process zone $l_{pz}$ [8], is of prime importance and the key to modelling fracture properties of quasi-brittle materials [9,10]. Considering this link, the height–height correlation function [11] or structure function [12] and correlation (auto-covariance) function are integrated in order to model the multiscale spectrum of fracture process zone (FPZ).

## 2. Materials and Methods

For this work, four different rock types including sandstone, marble and two granites with different grain sizes have been studied. The fractured surfaces are generated by performing notched semi-circular bending tests and under controlled conditions. The

topography of fractured surfaces, illustrated in Figure 1, has been reconstructed from 3D X-ray computed tomography data with a spatial resolution $d$ of about 16.5 µm. This method is superior to contact methods affected by the tip geometry of probes [13]. The average grain size of the studied rocks is quantified by some 2D slices of tomographic images and ranges from about 0.1 mm for the sandstone to around 1 mm for the marble. It is notable that before any post-processing on fractured surfaces, the global slope in the direction of crack propagation is removed by applying the gradient method such that the average height at the first and last lines of the height map in the propagation direction is the same.

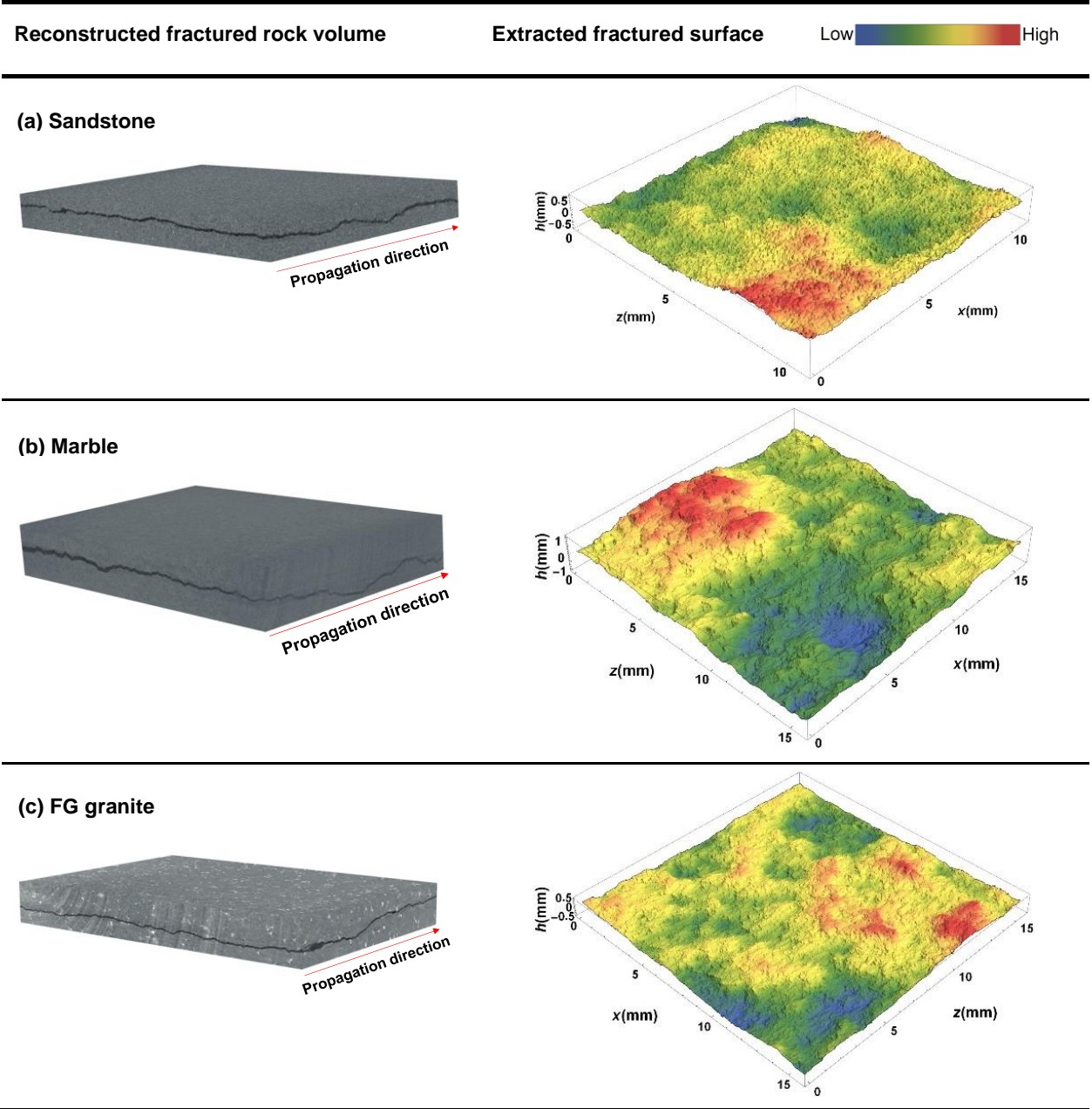

**Figure 1.** *Cont.*

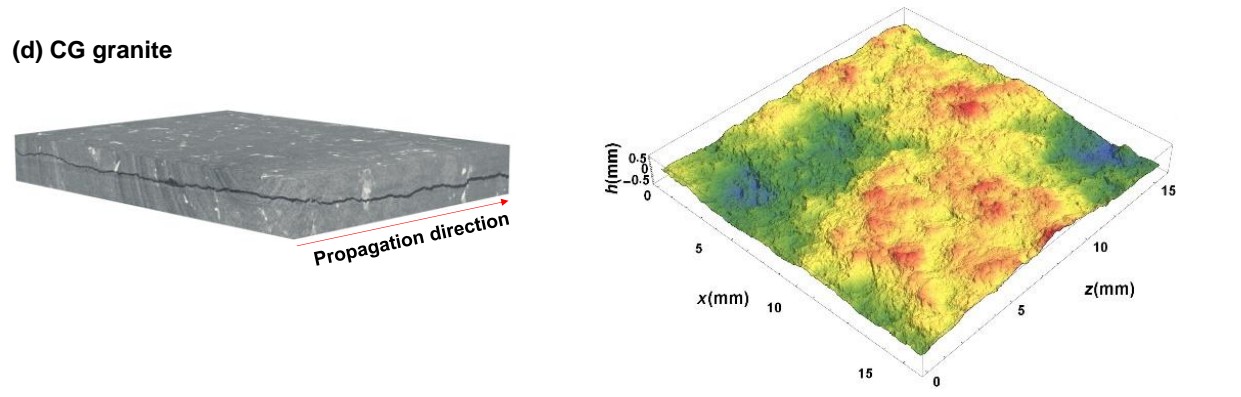

**(d) CG granite**

**Figure 1.** The 3D X-ray computed tomography images of the fractured area and topographic images of fractured surfaces of sandstone (**a**), marble (**b**) fine-grained granite (**c**), and coarse grained granite (**d**). The x-axis and z-axis correspond to the crack propagation direction and the crack front direction, respectively. The real length of reconstructed CT images is around 18 mm in the propagation direction.

Figure 2 is schematically illustrating the adopted methodology for modelling the multiscaling spectrum of FPZ of studied rocks. First, an initial guess of cross-over length $\xi$ is made by extrapolating the slope of the correlation function $\lambda_c$ calculated at some length scales $\epsilon$ to $C_\epsilon(\delta r) = 0$ at which $\delta r = \xi$. Then, the generalized Hurst exponent $H(q)$ is calculated for both monoaffine ($\delta r > \xi$) and multiaffine ($\delta r < \xi$) regimes. As opposed to monoaffine regime, at multiaffine one $H(q)$ nontrivially changes with $q$. This intermittency can be modelled with a perfect power law characterised by an exponent $\lambda_i$. If $\lambda_i = c * \lambda_c$, it means the initial estimate of $\xi$ was correct and both the true $\xi$ and $\lambda_c$ are correctly identified; otherwise, this estimate should be adjusted until $\lambda_i = c * \lambda_c$. $c$ is a constant, and based on experimental observations in this study, its value is considered to be related to the logarithmic mean $\log_e 10 / \log_{10} 10 \approx 2.3$. After the first iteration, the point of convergence of multiscale correlation functions and $\lambda_i$ are used to modify $\lambda_c$, and optimize the initial guess of $\xi$, accordingly.

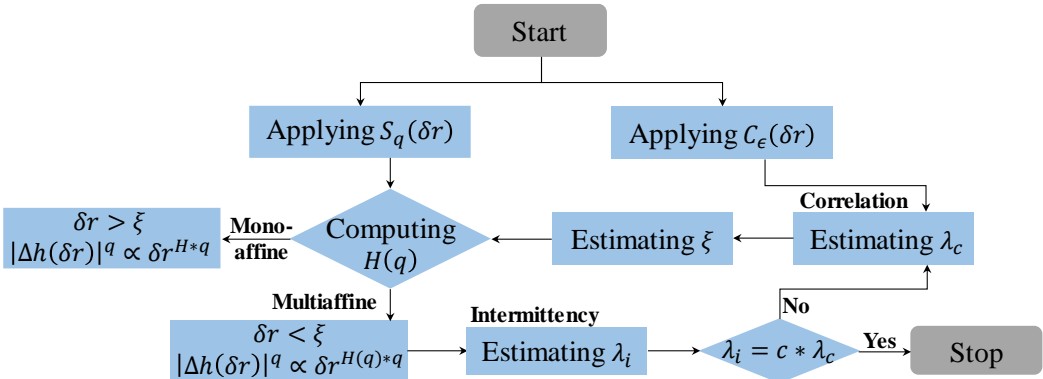

**Figure 2.** The outline of the adopted methodology for modelling the multiscaling spectrum of FPZ of studied rocks.

## 3. Results and Discussions

### 3.1. Roughness Correlation

These fractured surfaces exhibit anisotropic scaling properties. In order to identify the effective $\xi$ by means of the correlation function, height variations of fractured surfaces

$\Delta h(\epsilon)$ at some length scales $\epsilon$ are made statistically isotropic. $\omega_\epsilon(\mathbf{X})$ the operator has been used for such purpose [14]:

$$\omega_\epsilon(\mathbf{X}) = \frac{1}{2} \log\left(\left\langle \Delta h(\epsilon)^2 \right\rangle_\Theta\right) - \Omega_\epsilon, \tag{1}$$

where $\log(\Delta h)$ denotes the natural logarithm, whose base value is $e$. It measures the average height variations around each point, on a height field $h(\mathbf{X})$, with its neighbours over a circle of radius $\epsilon = d * p$ ($p = 1, 2, 3, 4, 5\ px$). $\epsilon$ is a product of spatial resolution and the number of pixels used for calculating $\Delta h(\epsilon)$; $\Omega_\epsilon$ is chosen such that the average of $\omega_\epsilon(\mathbf{X})$ overall $\mathbf{X}$ is zero. The pair length scale $\epsilon$ and direction n ($\epsilon$,n) = (16.5,8), (33,12), (49.5,16), (66,20), (82.5,28) is used to compute $\left\langle \Delta h(\epsilon)^2 \right\rangle_\Theta = \frac{1}{n} \sum_{k=0}^{n-1} \left[ h(x_i, z_j) - h\left(x_i + \epsilon * \cos(2\pi k / n), z_j + \epsilon * \sin(2\pi k / n)\right) \right]^2$ where $\Theta \in [0, 2\pi)\ rad$. For instance, at a length scale of $\epsilon = 16.5$ μm, 8 different data points are used to calculate the height difference for each point. $x_i$ and $z_j$ are the coordinates of a point on $h(\mathbf{X})$ in crack propagation and crack front directions, respectively. Figure 3 is showing the correlation functions $C_\epsilon(\delta r) = \left\langle \omega_\epsilon(\mathbf{X}) \omega_\epsilon(\mathbf{X} + \delta r) \right\rangle_\theta$ of $\omega_\epsilon$ fields averaged over 4 directions $\theta \in [0, \pi) = \pi k / 4\ rad$ ($k = 0, 1, 2, 3$). The best-fit line of correlation functions $C_\epsilon(\delta r) = -\lambda_c \log(\delta r / \xi) + \varepsilon$ passing through the point of convergence and the true $\xi$ with the slope $\lambda_c = \lambda_i / c$ for different rock types are shown (under the assumption that $\varepsilon = 0$). It is notable that $\lambda_c$ is dimensionless, since slopes of auto-correlation (dimensionless) and auto-covariance (dimensional) functions are the same. Indeed, $C_\epsilon(\delta r)$ indicates scale-dependency of the material's disorder since $\omega_\epsilon$ quantifies local height variations and removes global slopes caused by macroscopic and dynamic effects.

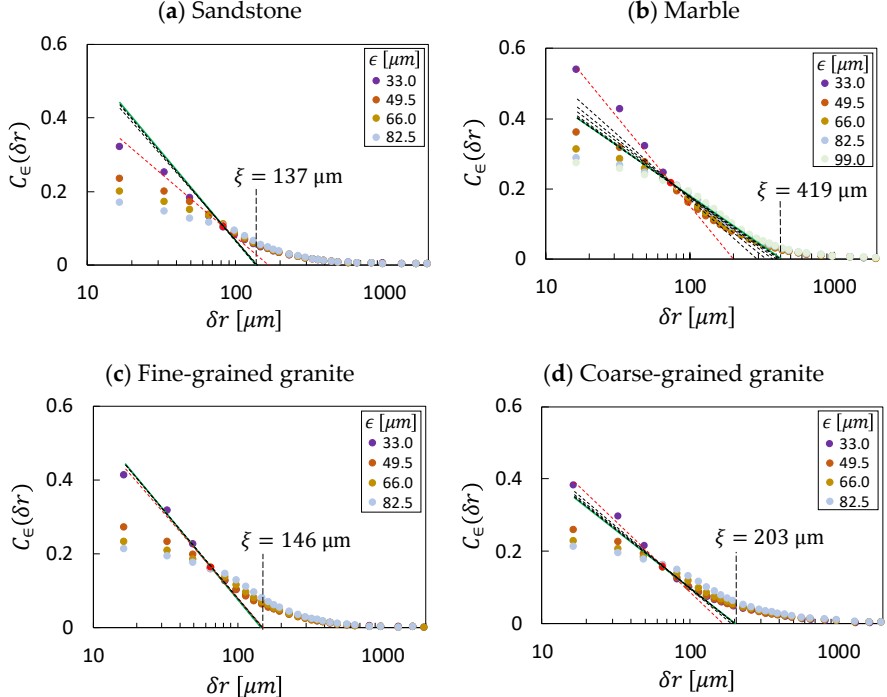

**Figure 3.** Spatial correlations of $\omega_\epsilon$ for sandstone (**a**), marble (**b**) fine-grained granite (**c**), and coarse grained granite (**d**). The correlations are represented for $\omega_\epsilon$ computed at different scales $\epsilon$. The true cutoff length $\xi$ is represented for each case. Dashed red lines are showing the initial guess for estimating $\lambda_c$; green solid lines are passing through the point of convergence (red points) and the true $\xi$ with the slope $\lambda_c = \lambda_i / c$; and black dashed lines are showing different iterations that are converging to the green lines.

### 3.2. Multifractality of Roughness

Following [15], the $q$th root of the $q$th moment of a statistical distribution of the height fluctuations $p(\Delta h)$, known as $q$th-order structure function $S_q$, is utilized to demonstrate

scale-dependency of the roughness of quasi-brittle fractured surfaces, and their transition from monoaffine to multiaffine surfaces at small enough separations:

$$S_q(\delta r) = \langle |\Delta h(\delta r)|^q \rangle^{1/q} = \langle |h(\mathbf{X} + \delta r) - h(\mathbf{X})|^q \rangle_{\mathbf{X}}^{1/q} \propto \delta r^{\zeta_q * 1/q}, \tag{2}$$

where angular brackets denote the ensemble average.

The concept of multiaffine fractals was introduced in the study of roughness analysis of growing surfaces [16]. FPZ is a multiaffine phenomenon showing different scaling properties in different directions $\theta \in [0, \pi)$ *rad* at different separations. However, it shall be shown that its average scaling properties over all directions show a multifractal process. In order to compute the multiscaling spectrum of the fracture process zone, $S_q(\theta, \delta r)$ has been calculated over admissible coordinates $(xo_i, zo_j)$. $xo_i$ and $zo_j$ are the coordinates of a point on a height field with zero average, $\langle h(\mathbf{X}) \rangle = 0$, in crack propagation and crack front directions, respectively. It is a very time-consuming process to compute many moments of structure function over all admissible coordinates and directions with small intervals for large data sets in order of million data points. $m = 12$ moments, $q = \{0.2, 0.4, 0.6, 0.8, 1, 1.5, 2, 2.5, 3, 4, 5, 6\}$, $n = 4$ directions, and $\theta = \pi k/n$ *rad* $(k = 0, 1, \dots, n-1)$ have been selected, which are accurate enough for analysing the intermittency of the studied rocks. Moreover, an interval $l = 2$ *px* for calculating $S_q(\theta, \delta r)$ is used. Thus, $\Delta h(\delta r)$ has been calculated between the following admissible coordinates $h\left(xo_{il}, zo_{jl} + \delta r\right) - h\left(xo_{il}, zo_{jl}\right)$ for selected moment and angle sets. It is notable that for a few smaller subsets $S_q(\theta, \delta r)$ have been calculated among both selected and all admissible coordinates $h(xo_i, zo_j + \delta r) - h(xo_i, zo_j)$ in different directions $n = \{4, 12, 36\}$, and differences between calculated average scaling properties $\langle S_q(\delta r)_\theta \rangle$ were negligible. Admissible coordinates would be different considering separation and direction. Maximum separation has always been about half of the minimum dimension of the studied surfaces to compute moments at different separations over enough and similar data points. Overall, 34 to 37 different separations $\delta r \in [16.5, 13200]$ µm have been selected considering the dimensions of studied surfaces. Multiscaling spectra of FPZ of studied rocks are presented in Figure 4.

### 3.3. Determining the Cut-Off Length

It is observed that heart rate time series display different multiscaling exponents because of long-range correlations [17]. The BDM model is proposed to provide a link between multifractality and long-range correlation of financial time series [18]:

$$H(q) = H - (q - 1)\frac{\lambda_c}{2}, \tag{3}$$

in which $H \equiv \zeta_1$, and $\lambda_c$ is the slope of the correlation function of a multifractal process. This means that $H(q) = H$ if there is no correlation, i.e., $\lambda_c = 0$, which is the case for monofractals. Otherwise, $H(q)$ decrease linearly with $q$ for $\lambda_c > 0$. The BDM model has been used to analyse the multifractal behaviour of FPZ [14]. However, this linear model could not provide a good estimation of such non-linear multifractal spectra of the studied rocks. In this study, an experimental power law is introduced that can exactly predict the statistically isotropic multiscaling spectrum of rock FPZ:

$$H(q) = H\left(q^{-\lambda_i}\right), \tag{4}$$

in which $\lambda_i$ is the intermittency (slope of the multiscaling spectrum) of a multifractal process. This model is derived from experimental observation of the intermittent behaviour of the multiscaling spectrum of FPZ of the studied rock materials.

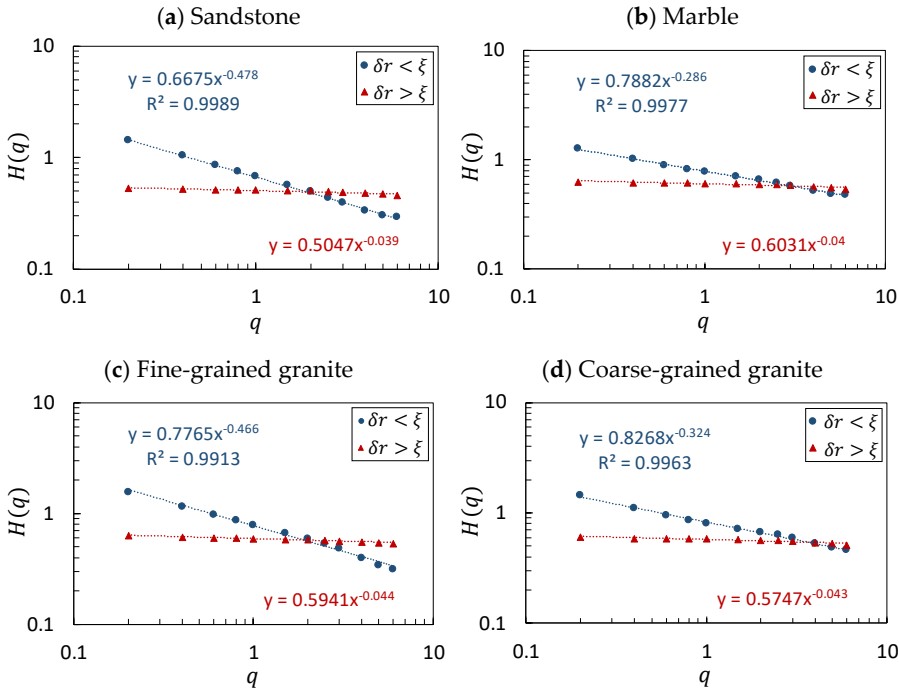

**Figure 4.** Multiscaling spectra of the rock fractured surfaces: sandstone (**a**), marble (**b**) fine-grained granite (**c**), and coarse grained granite (**d**). The spectra are computed both below (blue curve) and above (red curve) $\xi$. Both regimes are predicted with power laws; the best fit lines and their equations are represented with the same colour. Intermittency of multifractal regimes ($\delta r < \xi$) are showing perfect power laws with $R^2 \approx 1$ and some exponents between 0.25 and 0.5. Monofractal regimes ($\delta r > \xi$), however, show insignificant intermittency whose exponents are less than 0.05. Coefficients of presented equations represent $H(1) = \zeta_1/1$, which is sometimes considered as a Hurst exponent. Its values range from 0.5 to 0.6 for monofractal regimes, and from 0.65 to 0.85 for multifractal regimes.

By applying a log–log transformation, Equation (4) can be read as $\log H(q) = -\lambda_i \log q + \log H$. From Figure 4, it is indisputable that this log–log transformation is showing a perfect linear relationship that is unique to power functions. $\lambda_i$ computed from this relationship has been used to adjust $\lambda_c$ and optimize the initial guess of $\xi$. From Figure 3, it can be clearly seen that apparent correlation functions computed at different scales meet each other at a particular point, which is the true correlation function and reflects the intrinsic length of the material disorder. The coordinates of this point are the key to determine the true $\xi$ by modifying $\lambda_c$ based on computing $\lambda_i$. By repeating the loop shown in Figure 1 a few times (from 5 to 8 iterations in this study) both the intermittency of the multifractal process and the true cross-over length of the correlation function can be objectively identified. A comparison among experimental data, the BDM model and the proposed model is made in Figure 5, which shows the success of the proposed model in predicting multiscaling spectra of the rock fractured surfaces at small enough length scales.

If $\lambda_i = c * \lambda_c$, then $1 - \frac{\log(\zeta_q/H)}{\log(q)} = -c * \frac{C_\epsilon^*(\delta r) - \varepsilon}{\log(\delta r/\xi)}$, and this equation can be rearranged as follows:

$$\frac{\log(\zeta_q/H)}{\log(q)} - \frac{c * C_\epsilon^*(\delta r)}{\log(\delta r/\xi)} = 1 - \frac{c * \varepsilon}{\log(\delta r/\xi)}, \tag{5}$$

where $C_\epsilon^*(\delta r)$ is an auto-correlation function that is normalized by the variance of $\omega_\epsilon$ field. In the monoaffine regime, $\zeta_q/H = q$, there is no correlation $C_\epsilon^*(\delta r) \approx 0$, and Equation (5) is read as $1 - 0 = 1$. In the transition zone, the correlation from $\varepsilon$ at $\delta r = \xi$ would tend to zero as $\delta r$ tends to infinity. In the multiaffine regime, $\zeta_q/H = q^{1-\lambda_i}$, there is a correlation of $0 < C_\epsilon^*(\delta r) - \varepsilon < 1$, and Equation (5) read as $1 - \lambda_i + c * \lambda_c = 1$.

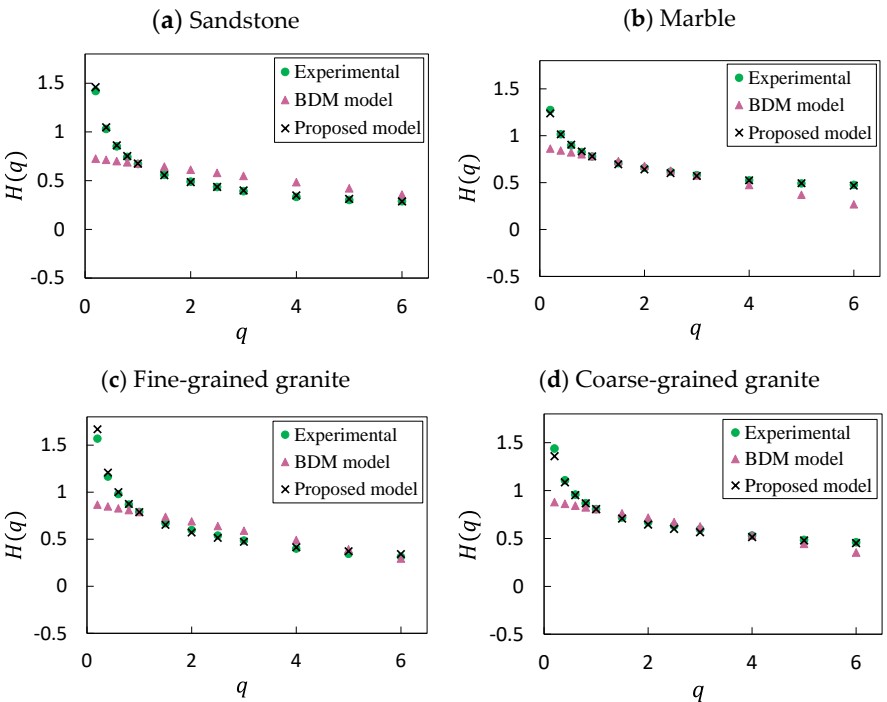

**Figure 5.** Experimental and predicted multiscaling spectra of the rock fractured surfaces: sandstone (**a**), marble (**b**) fine-grained granite (**c**), and coarse grained granite (**d**). Predicted spectra by the proposed model are very close to experimental ones.

### 3.4. Limitations and Future Work

The question that remains open here is as follows: what is the relationship between the slope of the correlation function and the intermittency? To answer this question, finding a meaningful link between the correlation function and the structure function is the first step. Two kinds of these links that can trigger some interesting future research are provided as follows.

The power spectrum, on the one hand, is the Fourier transform of the auto-covariance function of a wide-sense stationary random process (Wiener–Khinchin theorem). Height variations in the monofractal regime can be considered as a fractional Brownian motion with stationary increments and their spectral density $S(f)$ scales with the frequency $f$ as a power law: $S(f) \propto f^{-\beta}$ where $\beta = 2H(2) + 1$ [19]. On the other hand, the spectral density is proportional to the second moment of distribution of height variations $\left\langle |\Delta h(\delta r)|^2 \right\rangle \propto \delta r^\alpha$ where $\alpha = 2H(2) = \beta - 1$. For monofractals a relationship between $H(q) = H$ and $\beta$ can be defined as $D_q = q + 1 - H = q + \frac{3-\beta}{2}$. However, when it comes to multifractals with nonstationary increments, this link between the correlation function and structure function needs some modifications. In other words, like the generalized fractal dimension and Hurst exponent, a generalization for spectral analysis is required. Wigner–Ville spectral analysis is introduced as the unique generalized spectrum for spectral analysis of non-stationary processes [20]. There is also a statistical relationship between the structure function and correlation function: $\left\langle |\Delta h(\delta r)|^2 \right\rangle = 2\left(\sigma^2 - C_\epsilon(\delta r)\right)$, where $\sigma^2 = \left\langle h(\mathbf{X})^2 \right\rangle \approx \left\langle h(\mathbf{X} + \delta r)^2 \right\rangle$ is the variance of a height map. This equation is working very well for the second moment, but the question is how it can be expanded for other moments. Moreover, this relationship ignores the effect of local averaging on the height map (to compute $\omega_\epsilon$ field). Along with this theoretical issue, it seems the coefficient $c$ plays a very important role in understanding multifractal phenomena and can be used to estimate the crossover length of multiphase phenomena. Further measurements in different fields are required to have a better comprehension of the proposed model in this study. The outcomes of this research can pave the

way to deciphering multiscaling features of quasi-brittle fractured surfaces and model the roughness of the fractured rocks.

## 4. Conclusions

In this study, the intermittency of the roughness of fractured surfaces of different rock types is quantified. Based on the studied fractured surfaces the intermittency can be modelled by perfect power laws that can be further used to analyse multiscale properties of the roughness of fractured surfaces at length scales smaller than the cut-off length, an important topic in fluid mechanics. Moreover, a repetitive loop is introduced to determine the cut-off length, which is the length scale at which a transition from multi- to monofractality of the fracture surfaces can be observed, by making a connection between the slope of intermittency and slope of the correlation function of the same surface. Finally, some directions for future research are introduced that can further enhance our understanding of the multiscaling features of rough surfaces.

**Author Contributions:** Conceptualization, S.A. and M.K.; methodology S.A.; software, S.A.; validation, S.A. and M.K.; formal analysis, S.A.; investigation, S.A.; resources, S.A. and M.K.; data curation, S.A.; writing—original draft preparation, S.A. and M.K.; writing—review and editing, S.A. and M.K. All authors have read and agreed to the published version of the manuscript.

**Funding:** This study received no external funding.

**Data Availability Statement:** Data can be accessed by contacting to the corresponding author.

**Acknowledgments:** S.A. wishes to acknowledge the support from the Australian Government Research Training Program (RTP) Scholarship and the Monash International Tuition Scholarship (MITS). This research is supported by the Australian Synchrotron, and the MASSIVE HPC facility (www.massive.org.au).

**Conflicts of Interest:** The authors declare no conflict of interest.

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
