# Peer review of "Intermittency of Rock Fractured Surfaces: A Power Law"

_water, doi:10.3390/w14223662_

Round 1

Reviewer 1 Report

This paper introduced a power law to forecast the multiscaling spectrum of rock fracture process with the aid of CT scanning and image processing techniques. This work is interesting and important for roughness quantification. However, the authors need to make necessary improvement. My comments as follow:

1. Line 33, first occurrence symbol δr should be explained.

2. Line 50, how is the fracture of rock obtained? Is it natural or obtained through shear test? Are the four kinds of rock fractures obtained consistently?

3. Line 83-84, what is the specific meaning of variable n? for example, which direction is n=8? It’s worth mentioning.

4. Line 117-119, authors select a large number of representative parameters to reduce the calculation, are these parameters selected based on experience, and do they affect the selection of parameters for different rocks?

5. There are many variables and symbols, it is recommended to add variable annotation table.

6. It seems unreasonable that there is no conclusion in the manuscript. It is necessary for the author to supplement this part.

Author Response

Dear Reviewer,

Thanks for your consideration of our work. Your comments were insightful and helped authors improve the manuscript. The authors fully considered your views and revised the manuscript accordingly. All the modifications can be seen in the revised version.

Reviewer 2 Report

Dear authors, In our opinion, it is necessary to add information about the size of the grains of the rock. It is necessary to bring the ratio of the grain sizes of the studied rocks to the resolution of computed tomography, this would make it possible to verify the accuracy of the models obtained.

Author Response

Dear Reviewer

Thanks for your consideration of our work. Your comment were insightful and helped authors improve the manuscript. The authors fully considered your view and revised the manuscript accordingly. We have explained the grain size of all rocks are much higher than the resolution of the tomograph.

Round 2

Reviewer 1 Report

The authors have modified the manuscript and I think it can be published in this journal.